# Risk of carotid plaques according to triglyceride-glucose index stratified by thyroid function: A cross-sectional study

**Hye Jeong Kim** [1], **Seong Soon Kwon** [2], **Sang Joon Park**[1], **Dong Won Byun**[1], **Kyoil Suh**[1], **Myung Hi Yoo**[1], **Duk Won Bang**[2‡]*, **Hyeong Kyu Park**[1‡]*

**1** Division of Endocrinology and Metabolism, Department of Internal Medicine, Soonchunhyang University Hospital, Soonchunhyang University College of Medicine, Seoul, Korea, **2** Division of Cardiology, Department of Internal Medicine, Soonchunhyang University Seoul Hospital, Soonchunhyang University College of Medicine, Seoul, Korea

☯ These authors contributed equally to this work.
‡ HKP and DWB also contributed equally to this work as corresponding author.
* hkpark@schmc.ac.kr (HKP); schbdw@schmc.ac.kr (DWB)

**Data Availability Statement:** All relevant data are within the paper and its Supporting Information files.

## Abstract

### Background

Recent studies have indicated that the triglyceride-glucose (TyG) index or subclinical thyroid dysfunction is associated with carotid plaques, a predictor of cardiovascular disease risk. However, evidence for this association is limited and inconsistent. This study aimed to evaluate the risk of carotid plaques according to TyG index and thyroid function status in the general population.

### Methods

A total of 2,931 individuals who underwent carotid ultrasound as part of a comprehensive health examination at the Health Promotion Center of Soonchunhyang University Hospital were retrospectively reviewed. Based on the TyG index and thyroid function status, the participants were divided into six groups: LoTyG-SHyper (low TyG index with subclinical hyperthyroidism), LoTyG-Eu (low TyG index with euthyroidism), LoTyG-SHypo (low TyG index with subclinical hypothyroidism), HiTyG-SHyper (high TyG index with subclinical hyperthyroidism), HiTyG-Eu (high TyG index with euthyroidism), and HiTyG-SHypo (high TyG index with subclinical hypothyroidism). A multivariate logistic regression analysis was conducted to determine the risk of carotid plaques.

### Results

The proportion of participants with significant carotid plaques was significantly different among the six groups (p<0.001, p for trend<0.001). The odds ratio (OR) and 95% confidence interval (CI) for significant carotid plaques were significantly higher in the HiTyG-SHypo group than in the LoTyG-Eu group, even after adjusting for confounding variables including sex, age, smoking, obesity, hypertension and diabetes mellitus (OR 1.506, 95% CI 1.045–2.170, p = 0.028). The OR of significant carotid plaques was higher in the HiTyG-Eu

**Funding:** This study was supported by Soonchunhyang University Research Fund to Dr. Hye Jeong Kim.

**Competing interests:** The authors have no competing interests to disclose.

group than in the LoTyG-Eu group; however no associations were observed after additional adjustment for confounding variables.

## Conclusion

The TyG index and thyroid function status are important predictors of the risk of carotid plaques in healthy individuals. Early evaluation of carotid plaques may be necessary for subjects with high insulin resistance and subclinical hypothyroidism.

## Introduction

Carotid artery plaques have been studied as a surrogate marker of cardiovascular diseases (CVD), further stratifying at-risk individuals to help identify those most likely to benefit from aggressive medical therapy and lifestyle modifications [1–3]. Recent studies suggest that quantitative carotid plaque scores may be the best carotid ultrasound predictors of CVD risk [4, 5].

Insulin resistance is an independent risk factor for CVD [6]. The triglyceride-glucose (TyG) index is regarded as a valuable biomarker of insulin resistance [7]. Recently, several studies have reported that the TyG index is associated with carotid plaques in the general population [8–13].

Thyroid hormones are involved in energy homeostasis [14], lipid and glucose metabolism [15–17], and blood pressure (BP) [18]. Thyroid dysfunction is associated with an increased risk of CVD [19, 20]. Data on the role of subclinical thyroid dysfunction in the development of atherosclerosis are still inconclusive. Although some studies have found no association between subclinical thyroid dysfunction and carotid plaques [21, 22], other studies have identified subclinical thyroid dysfunction as a risk factor for carotid plaques [23, 24]. Previous studies have shown an association between thyroid function and insulin resistance measured using the TyG index [25, 26]. To the best of our knowledge, to date, no studies have addressed the differences in the risk of carotid plaques according to the TyG index and thyroid function status.

In the present study we aimed to evaluate the risk of carotid plaques according to the TyG index and thyroid function status in the general population.

## Subjects and methods

### Study population

This study retrospectively reviewed the data of 2,931 individuals who underwent carotid ultrasound as part of a comprehensive health examination at the Health Promotion Center of Soonchunhyang University Hospital between January 2016 and June 2018. Among these participants, 28 were excluded for abnormal free thyroxine (fT4) levels [fT4 <0.89 ng/dL (n = 9) and fT4 >1.78 ng/dL (n = 19)]. Finally, 2,903 subjects were eligible for analysis. All participants were categorized into six groups based on their TyG index and thyroid function status: LoTyG-SHyper (low TyG index with subclinical hyperthyroidism), LoTyG-Eu (low TyG index with euthyroidism), LoTyG-SHypo (low TyG index with subclinical hypothyroidism), HiTyG-SHyper (high TyG index with subclinical hyperthyroidism), HiTyG-Eu (high TyG index with euthyroidism) and HiTyG-SHypo (high TyG index with subclinical hypothyroidism).

The subjects' anthropometric data, laboratory test results, and coded answers to self-reported questionnaires were stored in the electronic medical records. The requirement for informed consent for this study was waived by the institutional review board because we only accessed the database for analysis purposes and did not access personal identifying information. The study protocol was approved by the Institutional Review Board of Soonchunhyang University Hospital (approval number: 2022-09-006).

## Clinical and laboratory measurements

Clinical variables for each individual were obtained from the medical records: sex, age, smoking, body mass index (BMI), systolic BP, diastolic BP, history of the medical disease (diabetes mellitus and hypertension) and use of medications (oral hypoglycemic agents, insulin and anti-hypertensive drugs).

Smoking status was evaluated using a questionnaire completed during an interview and participants were defined as current, former, or never smokers [27].

Height and weight were measured while the participants were wearing light clothing without shoes. BMI was calculated as the weight in kilograms divided by the square of height in meters (kg/m$^2$). BP was measured using an automatic manometer with subjects in a sitting position All BP measurements were taken in triplicate, and the mean of the second and third measured values was used in the analyses.

After overnight fasting, blood samples were drawn from the antecubital vein into vacuum tubes and subsequently analyzed at a central, certified laboratory at Soonchunhyang University Hospital. Triglycerides, high-density lipoprotein (HDL) cholesterol and fasting glucose levels were measured using Cobas 8000 C702 (Roche Diagnostics System, Switzerland). Hemoglobin A1c (HbA1c) was measured using an immunoturbidimetric assay with a Cobas Integra 800 automatic analyzer (Roche Diagnostics) with a reference value range of 4.4–6.4%. HbA1c measurements were standardized to the reference method in the Diabetes Control and Complications Trial and according to the National Glycohemoglobin Standardization Program standards. Serum thyroid-stimulating hormone (TSH) levels were measured using an immunoradiometric assay with a TSH-CTK-3 kit (DiaSorin SpA, Saluggia, Italy) with a laboratory reference range of 0.3–4.0 mIU/L. Serum fT4 levels were assessed using a radioimmunoassay with an FT4 RIA kit (Immunotech, Prague, Czech Republic), with a laboratory reference range of 0.89–1.78 ng/dL.

## Carotid ultrasound examination

A high-resolution B-mode ultrasound (EPIQ 5C or IE 33 USG systems; Philips, Andover, Massachusetts, USA) equipped with an 11 MHz linear array transducer was used to assess carotid artery plaques. The bilateral carotid arteries were scanned with the beam focused on the near and far walls of the distal 2 cm, or the common carotid artery proximal to its bifurcation. Both transverse and longitudinal images were obtained for extensively evaluated plaques. Carotid plaque presence was defined as focal abnormal wall thickness [intima-media thickness (IMT) >1.5 mm] or focal thickening of >50% of the surrounding IMT [28, 29]. To describe carotid plaque burden, the carotid plaque score was calculated as a total number of sites with plaques ranging from 0 to 6 (right- and left-sided common carotid artery, bifurcation and internal/external carotid artery) [30, 31]. All measurements were performed using the same device by the same experienced sonographer (SM Yoon, a registered diagnostic cardiac sonographer with 10 years of experience). The carotid ultrasound examination results were reviewed by two physicians (SS Kwon and DW Bang).

## Definitions

The TyG index was calculated as ln[fasting triglycerides (mg/dL) × fasting blood glucose (mg/dL)/2] [7]. By dividing the 50th percentile according to the TyG index values, a TyG index value <8.625 was defined as a low TyG index group and a TyG index value ≥8.625 was defined as a high TyG index group.

Euthyroidism was defined as serum TSH and fT4 levels within the normal reference range. Subclinical hyperthyroidism was defined as TSH levels <0.3 mIU/L and normal fT4 levels, and subclinical hypothyroidism was defined as TSH levels >4.0 mIU/L and normal fT4 levels.

A significant carotid plaque was defined as a carotid plaque score ≥2.

## Statistical analysis

Continuous variables are reported as medians with interquartile ranges, and categorical variables are presented as percentages (%). The demographic and biochemical characteristics of the study population with respect to the TyG index and thyroid function status were compared using the Mann-Whitney U-test or Kruskal-Wallis test for continuous variables and the χ2 test or Fisher's exact test (for small cell values) for categorical variables.

Logistic regression analyses were used to estimate the odds ratios (ORs) with 95% confidence intervals (CIs) for the risk of carotid plaques by location and significant carotid plaques. All p values and 95% CI for OR were corrected using Bonferroni's method due to multiple testing. Additional adjustments were made for confounding variables, such as sex, age (years), smoking (current smoker, ex-smoker or never smoker), obesity (BMI ≥25 kg/m$^2$ or BMI <25 kg/m$^2$), hypertension (BP ≥140/90 mmHg/antihypertensive medication or no) and diabetes mellitus (HbA1c ≥6.5%/antidiabetic medication or no).

All statistical analyses were performed using SPSS Statistics version 26.0 (IBM Corp., Chicago, IL, USA). Item analysis with a two-sided p value <0.05 was considered statistically significant.

## Results

The baseline clinical and biochemical characteristics of the 2,903 participants are summarized in Table 1. Of these, 2,213 (76%) were male, with a median age of 52.0 (26.0–87.0) years. The median values of TSH and fT4 were 2.03 (0.0–25.23) mIU/L and 1.31 (0.89–1.78) ng/dL, respectively.

Subjects with a high TyG index were older, male, more likely to be current smokers, and tended to have a history of hypertension or diabetes mellitus than those with a low TyG index. They also had higher BMI, BP, HDL cholesterol, triglycerides, fasting glucose, HbA1c, TyG index, C-reactive protein, carotid IMT and carotid plaque scores than those with a low TyG index. There were no significant differences in TSH and fT4 levels between the low TyG index and high TyG index groups. There were also no significant differences in TSH and fT4 levels between the LoTyG-SHyper and HiTyG-SHyper groups, LoTyG-Eu and HiTyG-Eu groups, and LoTyG-SHypo and HiTyG-SHypo groups.

In the low TyG index group, there were statistically significant differences in terms of age, sex, and proportion of current smokers according to thyroid function status (Table 2A). In the high TyG index group, there were statistically significant differences in terms of sex, proportion of current smokers, and total cholesterol levels according to thyroid function status (Table 2B).

The proportion of participants with carotid plaques by location showed a significant difference among the six groups in the common carotid artery (p<0.001), bifurcation (p = 0.003),

**Table 1. Baseline characteristics of participants with respect to the triglyceride-glucose index.**

| Variables | Low TyG index (n = 1457) | High TyG index (n = 1446) | p value | Overall (N = 2931) |
|---|---|---|---|---|
| Age (years) | 51.0 (43.0, 57.0) | 53.0 (74.0, 58.0) | <0.001 | 52.0 (46.0, 58.0) |
| Sex (male), n (%) | 946 (65%) | 1267 (88%) | <0.001 | 2213 (76%) |
| Current smoker, n (%) | 244 (17%) | 420 (29%) | <0.001 | 664 (23%) |
| Hypertension, n (%)[1] | 273 (19%) | 445 (31%) | <0.001 | 718 (25%) |
| Diabetes mellitus, n (%)[2] | 56 (4%) | 249 (18%) | <0.001 | 305 (10%) |
| BMI (kg/m$^2$) | 23.4 (21.5, 25.2) | 25.5 (23.8, 27.5) | <0.001 | 24.4 (22.6, 26.5) |
| Obesity, n (%)[3] | 398 (27%) | 833 (58%) | <0.001 | 1231 (42%) |
| Systolic BP (mmHg) | 119 (109, 129) | 126 (116, 135) | <0.001 | 123 (112, 132) |
| Diastolic BP (mmHg) | 73 (66, 80) | 78 (71, 85) | <0.001 | 75 (69, 82) |
| Total cholesterol (mg/dL) | 185 (161, 207) | 200 (177, 224) | <0.001 | 192 (168, 215) |
| HDL cholesterol (mg/dL) | | | | |
| Male | 58.0 (50.0, 67.0) | 48.0 (41.0, 55.0) | <0.001 | 52.0 (44.0, 61.0) |
| Female | 68.0 (59.0, 79.0) | 53.0 (45.0, 63.0) | <0.001 | 64.0 (54.0, 76.0) |
| Triglycerides (mg/dL) | 84 (66, 102) | 170 (137, 219) | <0.001 | 117 (84, 169) |
| Fasting glucose (mg/dL) | 88 (83, 94) | 97 (90, 108) | <0.001 | 92 (86, 101) |
| HbA1c (%) | 5.4 (5.2, 5.6) | 5.6 (5.3, 6.1) | <0.001 | 5.5 (5.2, 5.8) |
| TyG index[4] | 8.23 (7.98, 8.44) | 9.05 (8.81, 9.33) | <0.001 | 8.62 (5.24, 9.04) |
| TSH (mIU/L) | 2.06 (1.39, 3.05) | 1.99 (1.38, 2.83) | 0.113 | 2.03 (1.38, 2.98) |
| fT4 (ng/dL) | 1.30 (1.18, 1.42) | 1.32 (1.20, 1.42) | 0.089 | 1.31 (1.19, 1.42) |
| CRP (mg/L) | 0.04 (0.02, 0.08) | 0.07 (0.04, 1.14) | <0.001 | 0.05 (0.03, 1.11) |
| Carotid IMT (mm)[5] | 0.54 (0.48, 0.61) | 0.58 (0.52, 0.65) | <0.001 | 0.56 (0.50, 0.63) |
| Carotid plaque score[6] | 2.0 (0.0, 2.0) | 2.0 (1.0, 3.0) | <0.001 | 2.0 (1.0, 3.0) |

TyG, triglyceride-glucose; BMI, body mass index; BP, blood pressure; HDL, high-density lipoprotein; HbA1c, glycated hemoglobin; TSH, thyroid-stimulating hormone; fT4, free thyroxine; CRP, C-reactive protein; IMT, intima-media thickness.

Data are presented as numbers (percentage) or median (25th, 75th percentiles) as appropriate for the variable. Demographic and biochemical characteristics of the study population with respect to the triglyceride-glucose index were compared using $\chi^2$ test for categorical variables and the Mann-Whitney U-test for continuous variables.

[1] Blood pressure ≥140/90 mmHg or antihypertensive medication.

[2] Glycated hemoglobin ≥6.5% or antidiabetic medication.

[3] BMI ≥25 kg/m$^2$ according to the World Health Organization standards for Asians.

[4] Ln [fasting triglycerides (mg/dL) × fasting glucose (mg/dL)/2].

[5] Mean intima-media thickness of right and left carotid artery.

[6] The carotid plaque score ranges from 0 to 6, depending on the presence or absence of plaque in bilateral common carotid artery, bifurcation and internal/external carotid artery.

and internal carotid artery (p = 0.019) (Table 3). For the external carotid artery, there was no difference in the proportion of carotid plaques among the six groups.

In site-specific carotid plaque risk analysis, subjects in the LoTyG-SHypo, HiTyG-Eu and HiTyG-SHypo groups had a significantly higher risk of the common carotid artery and any site plaques than those in the LoTyG-Eu group. Subjects in the HiTyG-Eu and HiTyG-SHypo groups had a significantly higher risk of bifurcation plaques than those in the LoTyG-Eu group. Subjects in the HiTyG-Eu group had a higher risk of internal carotid artery plaques than those in the LoTyG-Eu group.

The proportion of subjects with significant carotid plaques was significantly different among the six groups and tended to increase significantly (Fig 1, p<0.001, p for trend<0.001).

We performed logistic regression analyses of the risk of significant carotid plaques among the subjects, using the LoTyG-Eu group as the reference category (Table 4).

**Table 2. Clinical and biochemical characteristics of participants according to thyroid function status.**

(A) Low TyG index.

| Variables | Low TyG index (n = 1457) | | | p value |
|---|---|---|---|---|
| | Subclinical hyperthyroidism (n = 24) | Euthyroidism (n = 1248) | Subclinical hypothyroidism (n = 185) | |
| Age (years) | 54.0 (47.0, 56.0)[ac] | 51.0 (43.0, 57.0)[a] | 53.0 (47.0, 59.0)[bc] | 0.001 |
| Sex (male), n (%) | 8 (33%) | 844 (68%) | 94 (51%) | <0.001 |
| Current smoker, n (%) | 3 (13%) | 224 (18%) | 17 (9%) | 0.003 |
| Hypertension, n (%)[1] | 2 (8%) | 233 (19%) | 38 (21%) | 0.349 |
| Diabetes mellitus, n (%)[2] | 0 (0%) | 46 (4%) | 10 (5%) | 0.322 |
| BMI (kg/m$^2$) | 23.9 (21.1, 25.0) | 23.4 (21.6, 25.2) | 23.3 (21.2, 25.1) | 0.735 |
| Obesity, n (%)[3] | 5 (21%) | 342 (27%) | 51 (28%) | 0.772 |
| Systolic BP (mmHg) | 121 (111, 128) | 119 (110, 129) | 118 (108, 128) | 0.220 |
| Diastolic BP (mmHg) | 71 (67, 76) | 73 (66, 80) | 72 (66, 78) | 0.177 |
| Total cholesterol (mg/dL) | 184 (163, 193) | 185 (161, 207) | 186 (161, 215) | 0.225 |
| HDL cholesterol (mg/dL) | | | | |
| Male | 54.0 (48.3, 62.0) | 58.0 (50.0, 67.8) | 56.5 (49.8, 64.3) | 0.357 |
| Female | 60.0 (51.8, 73.0) | 69.0 (59.0, 80.0) | 67.0 (60.0, 77.0) | 0.209 |
| Triglycerides (mg/dL) | 83 (71, 104) | 84 (66, 102) | 82 (67, 100) | 0.982 |
| Fasting glucose (mg/dL) | 90 (85, 94) | 88 (83, 94) | 88 (82, 93) | 0.707 |
| HbA1c (%) | 5.3 (5.2, 5.5) | 5.4 (5.2, 5.6) | 5.4 (5.2, 5.6) | 0.102 |
| TyG index[4] | 8.18 (8.10, 8.34) | 8.24 (7.98, 8.45) | 8.20 (7.96, 8.42) | 0.930 |
| TSH (mIU/L) | 0.05 (0.04, 0.16)[a] | 1.90 (1.35, 2.61)[b] | 5.04 (4.38, 5.83)[c] | <0.001 |
| fT4 (ng/dL) | 1.62 (1.40, 1.72)[a] | 1.30 (1.18, 1.41)[b] | 1.27 (1.14, 1.39)[c] | <0.001 |
| CRP (mg/L) | 0.4 (0.3, 1.0) | 0.4 (0.2, 0.8) | 0.4 (0.2, 0.7) | 0.705 |
| Carotid IMT (mm)[5] | 0.54 (0.47, 0.60) | 0.54 (0.48, 0.61) | 0.54 (0.49, 0.62) | 0.747 |
| Carotid plaque score[6] | 2.0 (0.0, 2.0)[ac] | 2.0 (0.0, 2.0)[a] | 2.0 (1.0, 3.0)[bc] | 0.045 |

(B) High TyG index.

| Variables | High TyG index (n = 1446) | | | p value |
|---|---|---|---|---|
| | Subclinical hyperthyroidism (n = 19) | Euthyroidism (n = 1267) | Subclinical hypothyroidism (n = 160) | |
| Age (years) | 50.0 (44.8, 55.3) | 53.0 (47.0, 58.0) | 54.0 (47.0, 60.0) | 0.293 |
| Sex (male), n (%) | 17 (89%) | 1121 (88%) | 129 (81%) | 0.017 |
| Current smoker, n (%) | 3 (16%) | 386 (30%) | 31 (20%) | 0.031 |
| Hypertension, n (%)[1] | 7 (37%) | 395 (31%) | 43 (27%) | 0.457 |
| Diabetes mellitus, n (%)[2] | 3 (16%) | 224 (18%) | 22 (14%) | 0.457 |
| BMI (kg/m$^2$) | 24.9 (23.5, 28.4) | 25.5 (23.9, 27.5) | 25.6 (23.6, 27.6) | 0.858 |
| Obesity, n (%)[3] | 9 (47%) | 735 (58%) | 89 (56%) | 0.561 |
| Systolic BP (mmHg) | 120 (116, 135) | 126 (116, 135) | 127 (117, 135) | 0.719 |
| Diastolic BP (mmHg) | 75 (67, 83) | 78 (71, 85) | 79 (70, 87) | 0.394 |
| Total cholesterol (mg/dL) | 196 (167, 208)[ac] | 199 (176, 223)[a] | 207 (186, 231)[bc] | 0.020 |
| HDL cholesterol (mg/dL) | | | | |
| Male | 49.0 (37.5, 56.0) | 48.0 (41.0, 55.0) | 48.0 (41.0, 56.0) | 0.992 |
| Female | 46.5 (38.0, 55.0) | 53.0 (46.0, 62.0) | 50.0 (45.5, 67.5) | 0.630 |
| Triglycerides (mg/dL) | 162 (139, 250) | 169 (136, 216) | 174 (147, 227) | 0.095 |
| Fasting glucose (mg/dL) | 97 (90, 113) | 98 (90, 108) | 96 (90, 109) | 0.523 |
| HbA1c (%) | 5.9 (5.3, 6.1) | 5.6 (5.3, 6.1) | 5.6 (5.3, 6.0) | 0.639 |
| TyG index[4] | 8.95 (8.75, 9.47) | 9.04 (8.80, 9.31) | 9.07 (8.84, 9.40) | 0.266 |
| TSH (mIU/L) | 0.08 (0.05, 0.12)[a] | 1.84 (1.32, 2.51)[b] | 5.11 (4.44, 6.57)[c] | <0.001 |
| fT4 (ng/dL) | 1.50 (1.43, 1.63)[a] | 1.32 (1.21, 1.42)[b] | 1.27 (1.13, 1.41)[c] | <0.001 |
| CRP (mg/L) | 0.8 (0.5, 2.6) | 0.7 (0.4, 1.4) | 0.7 (0.3, 1.4) | 0.499 |

*(Continued)*

**Table 2.** (Continued)

| | | | | |
|---|---|---|---|---|
| Carotid IMT (mm)[5] | 0.57 (0.55, 0.65) | 0.58 (0.52, 0.65) | 0.58 (0.52, 0.65) | 0.959 |
| Carotid plaque score[6] | 2.0 (1.0, 3.0) | 2.0 (1.0, 3.0) | 2.0 (1.0, 3.0) | 0.834 |

TyG, triglyceride-glucose; BMI, body mass index; BP, blood pressure; HDL, high-density lipoprotein; HbA1c, glycated hemoglobin; TSH, thyroid-stimulating hormone; fT4, free thyroxine; CRP, C-reactive protein; IMT, intima-media thickness.

Data are presented as numbers (percentage) or median (25th, 75th percentiles) as appropriate for the variable. Demographic and biochemical characteristics of the study population according to thyroid function status were compared using $\chi^2$ test for categorical variables and the Kruskal-Wallis test for continuous variables.

[a,b,c]The same letters indicate non-significant difference between groups based on Tukey's multiple comparison test.

[1]Blood pressure $\geq$140/90 mmHg or antihypertensive medication.

[2]Glycated hemoglobin $\geq$6.5% or antidiabetic medication.

[3]BMI $\geq$25 kg/m$^2$ according to the World Health Organization standards for Asians.

[4]Ln [fasting triglycerides (mg/dL) $\times$ fasting glucose (mg/dL)/2].

[5]Mean intima-media thickness of right and left carotid artery.

[6]The carotid plaque score ranges from 0 to 6, depending on the presence or absence of plaque in bilateral common carotid artery, bifurcation and internal/external carotid artery.

Subjects with HiTyG-Eu (OR 1.453, 95% CI 1.240–1.702, p<0.001) and HiTyG-SHypo (OR 1.893, 95% CI 1.338–2.680, p<0.001) had a significantly greater risk of significant carotid plaques than those in the LoTyG-Eu group. Additional adjustment were made for confounding variables, such as sex, age, smoking, obesity, hypertension and diabetes mellitus. Only HiTyG-SHypo remained a significant risk factor for significant carotid plaques, even after such adjustments (OR 1.506, 95% CI 1.045–2.170, p = 0.028).

## Discussion

In the present study, we found that the proportion of subjects with carotid plaques differed significantly according to insulin resistance and subclinical thyroid dysfunction. A high TyG index with subclinical hypothyroidism was associated with an increased probability of significant carotid plaques after adjusting for sex, age, smoking, obesity, hypertension and diabetes mellitus.

Recently, the TyG index has been considered an accessible and convenient measurement for estimating insulin resistance [32] and may be useful for predicting CVD events in clinical practice [33, 34]. The association between the TyG index and carotid plaques has been demonstrated in many studies [8–13]. The TyG index was positively associated with an elevated risk of carotid plaque prevalence [8, 9, 11, 13] and incidence [10, 12] in the general population. Consistent with previous studies, our results showed that a high TyG index was positively associated with carotid plaques and metabolic components such as obesity, high BP, low HDL cholesterol, high triglycerides and high glucose levels.

On the other hand, a recent study using a representative cohort of the Korean population reported that over hypothyroidism was correlated with an increased TyG index, and TSH was significantly correlated with the TyG index [26]. Another study using the same cohort found that the TyG index was associated with low-normal thyroid function in euthyroid Korean adults [25]. Therefore, we further investigated the risk of carotid plaques according to the TyG index, stratified by thyroid function status. The proportion of subjects with significant carotid plaques was significantly different according to insulin resistance and subclinical thyroid dysfunction. Subjects in the HiTyG-SHypo group had increased odds of the significant carotid plaques compared to those in the LoTyG-Eu group. This relationship remained statistically significant after adjusting for the major atherosclerotic risk factors.

**Table 3. Risk of carotid plaques by location according to the triglyceride-glucose index and thyroid function status group.**

| Location | LoTyG-SHyper (n = 24) | LoTyG-Eu (n = 1248) | LoTyG-SHypo (n = 185) | HiTyG-SHyper (n = 19) | HiTyG-Eu (n = 1267) | HiTyG-SHypo (n = 160) | p value |
|---|---|---|---|---|---|---|---|
| Common carotid artery | | | | | | | |
| n (%) | 3 (13%) | 262 (21%) | 54 (29%) | 2 (10%) | 403 (32%) | 51 (32%) | <0.001 |
| OR (95% CI) | 0.538 (0.159–1.816) | 1.000 | 1.551 (1.099–2.190)‡ | 0.443 (0.102–1.928) | 1.755 (1.466–2.102)* | 1.761 (1.229–2.522)† | |
| Bifurcation | | | | | | | |
| n (%) | 17 (71%) | 864 (69%) | 140 (76%) | 17 (89%) | 956 (75%) | 125 (78%) | 0.003 |
| OR (95% CI) | 1.079 (0.444–2.624) | 1.000 | 1.383 (0.968–1.975) | 3.778 (0.869–16.432) | 1.366 (1.146–1.628)* | 1.587 (1.071–2.353)‡ | |
| Internal carotid artery | | | | | | | |
| n (%) | 2 (8%) | 167 (13%) | 34 (18%) | 5 (26%) | 228 (18%) | 26 (16%) | 0.019 |
| OR (95% CI) | 0.588 (0.137–2.525) | 1.000 | 1.458 (0.971–2.188) | 2.312 (0.822–6.502) | 1.420 (1.143–1.765)† | 1.256 (0.800–1.971) | |
| External carotid artery | | | | | | | |
| n (%) | 0 (0%) | 34 (3%) | 2 (1%) | 1 (5%) | 36 (3%) | 3 (2%) | 0.619 |
| OR (95% CI) | NA | 1.000 | 0.390 (0.093–1.638) | 1.984 (0.257–15.291) | 1.044 (0.649–1.680) | 0.682 (0.207–2.247) | |
| Any location | | | | | | | |
| n (%) | 17 (71%) | 885 (71%) | 146 (79%) | 17 (89%) | 1003 (79%) | 129 (81%) | <0.001 |
| OR (95% CI) | 0.996 (0.410–2.422) | 1.000 | 1.536 (1.057–2.231)‡ | 3.486 (0.801–15.167) | 1.558 (1.298–1.870)* | 1.707 (1.132–2.574)‡ | |

LoTyG-SHyper, low triglyceride-glucose index with subclinical hyperthyroidism; LoTyG-Eu, low triglyceride-glucose index with euthyroidism; LoTyG-SHypo, low triglyceride-glucose index with subclinical hypothyroidism; HiTyG-SHyper, high triglyceride-glucose index with subclinical hyperthyroidism; HiTyG-Eu, high triglyceride-glucose index with euthyroidism; HiTyG-SHypo, high triglyceride-glucose index with subclinical hypothyroidism; OR, odds ratio; CI, confidence interval. OR and 95% CI for risk of carotid plaque by location according to the triglyceride-glucose index and thyroid function status group were estimated using logistic regression models.

*p<0.001,

†0.001≤p<0.01,

‡0.01≤p<0.05.

Several studies have attempted to link subclinical thyroid dysfunction with carotid plaques [21–24]. A small case-control study in Macedonia reported that subclinical hypothyroidism was associated with carotid plaques, independent of the risk factors for atherosclerosis [24], which supports some of our data. A population-based study in Germany demonstrated that subjects with decreased serum TSH levels had increased odds for carotid plaques compared to those with normal serum TSH levels, but there was no associations between elevated serum TSH levels and carotid plaques [23]. Another population-based cross-sectional study in Italy found no correlation between subclinical thyroid dysfunction and carotid plaques [21]. Recently, Kim et al. examined the association between subclinical thyroid dysfunction and carotid plaques in a cross-sectional and longitudinal assessment of Korean healthy individuals [22]. At baseline, carotid plaques were more prevalent in the subclinical hypothyroidism group than in the euthyroidism group [22]. However, the effect of subclinical hypothyroidism on the cumulative incidence of new carotid plaques was not significantly different with follow-up time [22]. These inconsistent findings [21–24] are possibly due to regional differences in iodine intake, which may affect thyroid function, and differences in the study design and study population. In addition, since insulin resistance was not considered in previous studies [21–24], additional studies to validate our findings are needed to determine whether the risk of carotid plaques differs according to subclinical thyroid dysfunction.

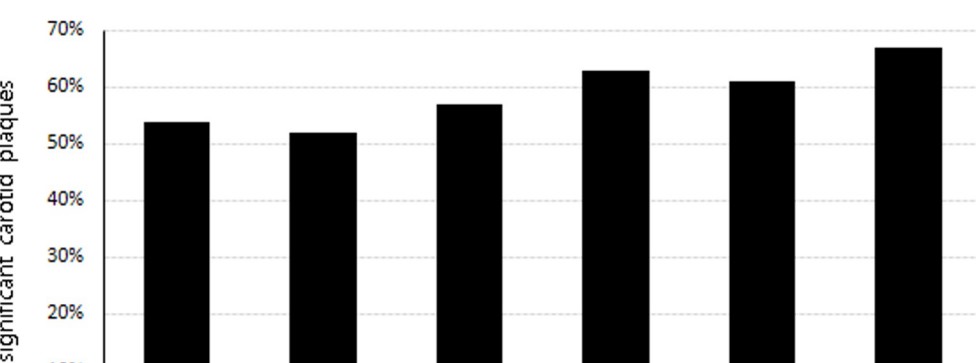

**Fig 1. Proportion of significant carotid plaques according to triglyceride-glucose index and thyroid function status.**

The biological mechanisms that could link a high TyG index and subclinical hypothyroidism with carotid plaques within our study population remain to be elucidated. In terms of insulin resistance, high TyG index and subclinical hypothyroidism appear to be associated with carotid plaques. The TyG index is a reliable marker of insulin resistance [7], and subclinical hypothyroidism is known to exhibit insulin resistance similar to overt hypothyroidism [35]. Insulin resistance can play an important role in the development of atherosclerosis via several mechanisms, including endothelial cell dysfunction [36, 37], proliferation and migration of vascular smooth muscle cells [38], modification of the synthesis and release of lipoproteins, inflammation, and reactive oxygen species formation [39, 40]. Regarding thyroid hormones themselves, subclinical hypothyroidism may be associated with atherosclerosis, as TSH directly stimulates hepatic gluconeogenesis [41] and cholesterol synthesis [42, 43], and leptin secretion in adipocytes [44]. In addition, thyroid hormone therapy has been shown to

**Table 4. Odds ratio (OR) and 95% confidence intervals (CI) for risk of significant carotid plaques based on the triglyceride-glucose index and thyroid function status.**

| | LoTyG-SHyper (n = 24) | LoTyG-Eu (n = 1248) | LoTyG-SHypo (n = 185) | HiTyG-SHyper (n = 19) | HiTyG-Eu (n = 1267) | HiTyG-SHypo (n = 160) |
|---|---|---|---|---|---|---|
| Significant carotid plaques, n (%) | 13 (54%) | 644 (52%) | 105 (57%) | 12 (63%) | 770 (61%) | 107 (67%) |
| Unadjusted | 1.108 (0.493–2.493) | 1.000 | 1.231 (0.902–1.681) | 1.608 (0.629–4.111) | 1.453 (1.240–1.702)* | 1.893 (1.338–2.680)* |
| Model 1 | 1.721 (0.744–3.982) | 1.000 | 1.277 (0.915–1.782) | 1.432 (0.548–3.747) | 1.226 (1.035–1.435)† | 1.548 (1.078–2.224)† |
| Model 2 | 1.725 (0.746–3.990) | 1.000 | 1.273 (0.911–1.777) | 1.341 (0.512–3.514) | 1.164 (0.972–1.393) | 1.506 (1.045–2.170)† |

LoTyG-SHyper, low triglyceride-glucose index with subclinical hyperthyroidism; LoTyG-Eu, low triglyceride-glucose index with euthyroidism; LoTyG-SHypo, low triglyceride-glucose index with subclinical hypothyroidism; HiTyG-SHyper, high triglyceride-glucose index with subclinical hyperthyroidism; HiTyG-Eu, high triglyceride-glucose index with euthyroidism; HiTyG-SHypo, high triglyceride-glucose index with subclinical hypothyroidism.

Model 1 adjusted for gender and age; Model 2 adjusted for gender, age, smoking, obesity, hypertension and diabetes mellitus.

OR and 95% CI for development of metabolic syndrome were estimated using logistic regression models.

*$p < 0.001$,

†$0.01 \leq p < 0.05$.

be effective in preventing the progression of atherosclerosis [45, 46] and contributing to endothelial-dependent vasodilation [47].

Despite the strength of large samples and control of extensive data on insulin resistance and several potential confounding factors, this study has some limitations. Due to its cross-sectional and retrospective nature, causal inferences could not be drawn. This study also lacked information on other exposures, including alcohol intake, physical activity, and medication with anti-thyroid drugs, thyroid hormones, or statins. Therefore, we cannot rule out the possibility of residual confounding variables for some measured and unmeasured factors. As subclinical thyroid dysfunction may be temporary, repeated measurement of thyroid function could provide a reliable result. A single measurement of thyroid function may have resulted in the inclusion of transient subclinical thyroid dysfunction. Therefore, additional prospective longitudinal studies are required.

In conclusion, our finding revealed a high TyG index with subclinical hypothyroidism is associated with an increased risk of carotid plaques in the general population. Early recognition of clinically useful biomarkers for carotid plaques could help identify subjects at high risk of CVD who may benefit from earlier medical treatment and lifestyle modifications.

## Supporting information

**S1 File.**
(XLSX)

## Author Contributions

**Conceptualization:** Hye Jeong Kim, Seong Soon Kwon, Duk Won Bang, Hyeong Kyu Park.

**Data curation:** Hye Jeong Kim, Seong Soon Kwon, Sang Joon Park, Dong Won Byun, Kyoil Suh, Myung Hi Yoo, Duk Won Bang.

**Formal analysis:** Hye Jeong Kim, Sang Joon Park, Dong Won Byun, Kyoil Suh, Myung Hi Yoo.

**Funding acquisition:** Hye Jeong Kim.

**Investigation:** Hye Jeong Kim.

**Methodology:** Hye Jeong Kim, Sang Joon Park, Dong Won Byun, Kyoil Suh, Myung Hi Yoo.

**Resources:** Seong Soon Kwon.

**Supervision:** Duk Won Bang, Hyeong Kyu Park.

**Writing – original draft:** Hye Jeong Kim.

**Writing – review & editing:** Seong Soon Kwon, Duk Won Bang, Hyeong Kyu Park.

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
