## [Decision Letter · Decision Letter 0]

10 Oct 2022

PONE-D-22-26387Risk of Carotid Plaques According to Triglyceride-Glucose Index Stratified by Thyroid FunctionPLOS ONE

Dear Dr. Kim,

Thank you for submitting your manuscript to PLOS ONE. After careful consideration, we feel that it has merit but does not fully meet PLOS ONE’s publication criteria as it currently stands. Therefore, we invite you to submit a revised version of the manuscript that addresses the points raised during the review process.

ACADEMIC EDITOR: Based on the publication criteria of PLOS ONE, I recommend minor revision of the issues that mentioned by the reviewers.==============================

We look forward to receiving your revised manuscript.

Kind regards,

Gulali Aktas

Academic Editor

PLOS ONE

Journal Requirements:

"This study was supported by Soonchunhyang University Research Fund to HJK."

"The authors have no competing interests to disclose."

Reviewers' comments:

Reviewer's Responses to Questions

**Comments to the Author**

1. Is the manuscript technically sound, and do the data support the conclusions?

Reviewer #1: Yes

Reviewer #2: Yes

2. Has the statistical analysis been performed appropriately and rigorously? 

Reviewer #1: Yes

Reviewer #2: Yes

3. Have the authors made all data underlying the findings in their manuscript fully available?

Reviewer #1: Yes

Reviewer #2: Yes

4. Is the manuscript presented in an intelligible fashion and written in standard English?

Reviewer #1: Yes

Reviewer #2: Yes

5. Review Comments to the Author

Reviewer #1: This study assessed the risk of carotid plaques according to TyG index and thyroid function status in the general population. This work is important and well-structured and the results support the authors ‘conclusions. Statistical procedures seemed steady. Confounding factors were excluded to conduct their results. These are the strengths of their study. I have some remarks aimed at improving the readability and presentation of the data.

1. The type of study (cross-sectional) is better to be mentioned in the title.

2. Inclusion and exclusion criteria should be added.

3. Please clarify statistical analysis section in more detail. For example, method of controlling confounders in this study.

4. In discussion section, although the results of the present study have been well compared with the findings of previous studies, the reason for discrepancy between the results of the current study and previous works should be discussed a little.

5. Authors need to add the strengths of this study before the paragraph explaining study limitations.

6. Please check the concordance of the tenses used throughout the text.

Reviewer #2: In this cross-sectional study, the authors aimed to evaluate the risk of carotid plaques

according to triglyceride-glucose index and thyroid function status in the general population.

The manuscript is written well, the text is fluent and easy to follow and, the statistical

analyzes are very satisfying. However, in this study; there are several minor points that can be

revised.

Minor points:

1. In the introduction section, the last sentence of the third paragraph, it is mentioned that

there are few studies in the literature on this topic discussed by the authors in the present

study. Please provide reference for this sentence.

2. Table 1 is very crowded and not easily understood as it is. It would be more appropriate to

present Table 1 as two new tables according to the low and high TyG index. Also, post-hoc

analyzes should be made for the pairwise comparisons after Kruskal-Wallis tests and this

statistical data added to the table footnotes.

3. According to current literature data, it is clear that high triglyceride-glucose index, presence

of subclinical hypothyroidism and insulin resistance increase cardiometabolic risk. Although

this study is original and comprehensive in terms of methodology and design, it is clear that

predictable results accordance with the literature will be obtained.

6. PLOS authors have the option to publish the peer review history of their article (what does this mean?). If published, this will include your full peer review and any attached files.

Reviewer #1: **Yes: **Helda Tutunchi

Reviewer #2: **Yes: **Mustafa Sait Gönen

---

## [Author Response · Author response to Decision Letter 0]

23 Nov 2022

Response to the Reviewer #1’s comments

We would like to express our sincere thanks for the careful review and helpful comments that have been of great assistance in improving this manuscript.

1. The type of study (cross-sectional) is better to be mentioned in the title.

Answer) Thank you for your valuable comments. We corrected the title as below.

Corrections) Title

Risk of Carotid Plaques According to Triglyceride-Glucose Index Stratified by Thyroid Function: a Cross-sectional Study

2. Inclusion and exclusion criteria should be added.

Answer) All participants who underwent carotid ultrasound underwent a health examination consisting of a series of checkup items including blood laboratory tests. And, if there were any items that were not written in the self-report questionnaire, they were asked to fill out the questionnaire again. So there were no participants to be excluded due to missing data. In addition, variables with high correlation with carotid plaque, such as hypertension or diabetes, were not excluded and were adjusted as confounding factors. However, because the study subject was limited to subclinical thyroid dysfunction and normal thyroid function, cases with abnormal fT4 results were excluded. We described the inclusion and exclusion criteria in the original manuscript as below.

In the original manuscript) Subject and methods section, Page 5, Line 55-60

This study retrospectively reviewed the data of 2,931 individuals who underwent carotid ultrasound as part of a comprehensive health examination at the Health Promotion Center of Soonchunhyang University Hospital between January 2016 and June 2018. Among these participants, 28 were excluded for abnormal free thyroxine (fT4) levels [fT4 <0.89 ng/dL (n=9) and fT4 >1.78 ng/dL (n=19)]. Finally, 2,903 subjects were eligible for analysis.

3. Please clarify statistical analysis section in more detail. For example, method of controlling confounders in this study.

Answer) Thank you for helpful comments. As you noted, the confounding variables were described in more detail as follows.

Corrections) Subject and methods section, Page 8, Line 137-141

Additional adjustments were made for confounding variables, such as sex, age (years), smoking (current smoker, ex-smoker or never smoker), obesity (BMI ≥25 kg/m2 or BMI <25 kg/m2), hypertension (BP ≥140/90 mmHg/antihypertensive medication or no) and diabetes mellitus (HbA1c ≥6.5%/antidiabetic medication or no).

4. In discussion section, although the results of the present study have been well compared with the findings of previous studies, the reason for discrepancy between the results of the current study and previous works should be discussed a little.

Answer) Thank you for the valuable comments. In discussion section, we described the region and population in which the study was conducted and corrected incomplete part about the reasons for discrepancy among studies.

Corrections) Discussion section, Page 18~19, Line 288-313

Therefore, we further investigated the risk of carotid plaques according to the TyG index, stratified by thyroid function status. The proportion of subjects with significant carotid plaques was significantly different according to insulin resistance and subclinical thyroid dysfunction. Subjects in the HiTyG-SHypo group had increased odds of the significant carotid plaques compared to those in the LoTyG-Eu group. This relationship remained statistically significant after adjusting for the major atherosclerotic risk factors.

Several studies have attempted to link subclinical thyroid dysfunction with carotid plaques [21-24]. A small case-control study in Macedonia reported that subclinical hypothyroidism was associated with carotid plaques, independent of the risk factors for atherosclerosis [24], which supports some of our data. A population-based study in Germany demonstrated that subjects with decreased serum TSH levels had increased odds for carotid plaques compared to those with normal serum TSH levels, but there was no associations between elevated serum TSH levels and carotid plaques [23]. Another population-based cross-sectional study in Italy found no correlation between subclinical thyroid dysfunction and carotid plaques [21]. Recently, Kim et al. examined the association between subclinical thyroid dysfunction and carotid plaques in a cross-sectional and longitudinal assessment of Korean healthy individuals [22]. At baseline, carotid plaques were more prevalent in the subclinical hypothyroidism group than in the euthyroidism group [22]. However, the effect of subclinical hypothyroidism on the cumulative incidence of new carotid plaques was not significantly different with follow-up time [22]. These inconsistent findings [21-24] are possibly due to regional differences in iodine intake, which may affect thyroid function, and differences in the study design and study population. In addition, since insulin resistance was not considered in previous studies [21-24], additional studies to validate our findings are needed to determine whether the risk of carotid plaques differs according to subclinical thyroid dysfunction.

5. Authors need to add the strengths of this study before the paragraph explaining study limitations.

Answer) Regarding the strength of our study, we described as below. Thank you for valuable comments.

Corrections) Discussion section, Page 19, Line 329-330

Despite the strength of large samples and control of extensive data on insulin resistance and several potential confounding factors, this study has some limitations.

6. Please check the concordance of the tenses used throughout the text.

Answer) Since our native language is not English, the manuscript has been proofread by a native English speaker and the certificate is attached as a supplementary file. Thank you very much.

 

Response to the Reviewer #2’s comments

We would like to express our sincere thanks for the careful review and helpful comments that have been of great assistance in improving this manuscript.

1. In the introduction section, the last sentence of the third paragraph, it is mentioned that there are few studies in the literature on this topic discussed by the authors in the present study. Please provide reference for this sentence.

Answer) To date, no papers have considered both TyG index and thyroid function. I used the word 'few' to emphasize 'no', but I realized that the English expression was wrong and corrected as below. Thank you for detailed comments.

Corrections) Introduction section, Page 4, Line 47-49

To the best of our knowledge, to date, no studies have addressed the differences in the risk of carotid plaques according to the TyG index and thyroid function status.

2. Table 1 is very crowded and not easily understood as it is. It would be more appropriate to present Table 1 as two new tables according to the low and high TyG index. Also, post-hoc analyzes should be made for the pairwise comparisons after Kruskal-Wallis tests and this statistical data added to the table footnotes.

Answer) Thank you for helpful comments. The table 1 was very crowded because it not only compared the low TyG and high TyG groups as a whole, but also analyzed according to thyroid function status in each TyG index group. So, Table 1 was changed to show the comparison between low TyG and high TyG groups, and Table 2 shows the comparison according to thyroid function status in each TyG index group. In addition, post-hoc analyses using Tukey’s multiple comparison test were made and added to the table footnotes.

Corrections) Table 1 and Table 2

3. According to current literature data, it is clear that high triglyceride-glucose index, presence of subclinical hypothyroidism and insulin resistance increase cardiometabolic risk. Although this study is original and comprehensive in terms of methodology and design, it is clear that predictable results accordance with the literature will be obtained.

Answer) We totally agree with your comments. Thank you very much for the detailed review and thoughtful comments.

---

## [Decision Letter · Decision Letter 1]

8 Dec 2022

Risk of Carotid Plaques According to Triglyceride-Glucose Index Stratified by Thyroid Function: a Cross-sectional Study

PONE-D-22-26387R1

Dear Dr. Kim,

We’re pleased to inform you that your manuscript has been judged scientifically suitable for publication and will be formally accepted for publication once it meets all outstanding technical requirements.

Kind regards,

Gulali Aktas

Academic Editor

PLOS ONE

Additional Editor Comments (optional):

Authors properly revised the manuscript. It is suitable now for publication in the journal.

Reviewers' comments:

Reviewer's Responses to Questions

**Comments to the Author**

1. If the authors have adequately addressed your comments raised in a previous round of review and you feel that this manuscript is now acceptable for publication, you may indicate that here to bypass the “Comments to the Author” section, enter your conflict of interest statement in the “Confidential to Editor” section, and submit your "Accept" recommendation.

Reviewer #1: All comments have been addressed

Reviewer #2: All comments have been addressed

2. Is the manuscript technically sound, and do the data support the conclusions?

Reviewer #1: Yes

Reviewer #2: Yes

3. Has the statistical analysis been performed appropriately and rigorously? 

Reviewer #1: Yes

Reviewer #2: Yes

4. Have the authors made all data underlying the findings in their manuscript fully available?

Reviewer #1: Yes

Reviewer #2: Yes

5. Is the manuscript presented in an intelligible fashion and written in standard English?

Reviewer #1: Yes

Reviewer #2: Yes

6. Review Comments to the Author

Reviewer #1: In the study entitled "Risk of Carotid Plaques According to Triglyceride-Glucose Index Stratified by Thyroid

Function: a Cross-sectional Study" all comments have been addressed carefully by the authors.

Reviewer #2: In line with our suggustions, the authors made the necessary corrections.As a result of the changes,the article has reached a better place.The manuscript can be accepted in this revised form.

7. PLOS authors have the option to publish the peer review history of their article (what does this mean?). If published, this will include your full peer review and any attached files.

Reviewer #1: **Yes: **Dr. Helda Tutunchi

Reviewer #2: **Yes: **Mustafa Sait Gönen

---

## [Editor Report · Acceptance letter]

19 Dec 2022

PONE-D-22-26387R1 

Risk of Carotid Plaques According to Triglyceride-Glucose Index Stratified by Thyroid Function: a Cross-sectional Study 

Dear Dr. Kim:

I'm pleased to inform you that your manuscript has been deemed suitable for publication in PLOS ONE. Congratulations! Your manuscript is now with our production department. 

Kind regards, 

on behalf of

Professor Gulali Aktas 

Academic Editor

PLOS ONE